META-RESEARCH

# Gender inequalities among authors who contributed equally

**Abstract** We analyzed 2898 scientific papers published between 1995 and 2017 in which two or more authors shared the first author position. For papers in which the first and second authors made equal contributions, mixed-gender combinations were most frequent, followed by male-male and then female-female author combinations. For mixed-gender combinations, more male authors were in the first position, although the disparity decreased over time. For papers in which three or more authors made equal contributions, there were more male authors than female authors in the first position and more all-male than all-female author combinations. The gender inequalities observed among authors who made equal contributions are not consistent with random or alphabetical ordering of authors. These results raise concerns about female authors not receiving proper credit for publications and suggest a need for journals to request clarity on the method used to decide author order among those who contributed equally.
DOI: https://doi.org/10.7554/eLife.36399.001

**NICHOLE A BRODERICK[†]\* AND ARTURO CASADEVALL[†]\***

**\*For correspondence:** nichole.
broderick@uconn.edu (NAB);
acasade1@jhu.edu (AC)

[†]These authors contributed
equally to this work; author order
was determined both
alphabetically and in order of
increasing seniority.

**Competing interests:** The
authors declare that no
competing interests exist.

**Reviewing editor:** Peter A
Rodgers, eLife, United Kingdom

## Introduction

In recent decades, the number of authors per publication has increased steadily (*National Library of Medicine, 2018*; *Colledge et al., 2013*). The causes for this trend include the higher production of scientific information by research teams (*Wuchty et al., 2007*) and an increase in the data content of published papers (*Cordero et al., 2016*; *Fontanarosa et al., 2017*), which in turn usually requires contributions by additional scientists. This increase in the number of authors per article has raised questions about credit allocation. Author order in an article byline is the major mechanism for assigning credit when there is more than one author. In the biomedical literature, the first author is usually the individual who has done most of the work and that individual traditionally receives most credit for the publication. This in turn has resulted in an increase of authors claiming equal credit in author byline positions, which has posed vexing questions as to how credit should be apportioned (*Tscharntke et al., 2007*; *Moustafa, 2016*).

Papers listing equal contributions comprised less than 1% of publications in 2000, but by 2009 this figure had increased to between 1.0% and 8.6%, depending on the journal (*Akhabue and Lautenbach, 2010*). For the journal Gastroenterology, for example, 21% of the papers published in 2011 and 2012 indicated two or more authors contributing equally (*Dubnansky and Omary, 2012*). Hence, the inclusion of statements of equal contribution by two or more authors is an increasingly common mechanism for sharing credit as the size of research teams increase in the biomedical sciences.

The shared authorship phenomenon is an important issue to study because the ability of junior investigators to publish first author papers is usually a necessary step for securing positions, acquiring funding, and receiving credit. To date very little scholarly work has been done to understand the mechanisms used in sharing credit allocation. In particular, we were interested in trends involving the sharing of equal contributions among authors differing in gender,

since inequities in distribution could translate into differences in gender recognition for scientific accomplishment. Numerous studies have documented underrepresentation of women in academic faculty and in scientific positions, especially at the more senior ranks (*Awad et al., 2017*; *Sassler et al., 2017*; *John et al., 2016*; *Hill et al., 2015*). Although the mechanisms for these trends are complex, one possibility is that they receive less credit for their scientific work (*John et al., 2016*). Several studies have documented gender differences in the frequency of first authors, with women less likely to occupy the first position (*Fishman et al., 2017*; *Bonham and Stefan, 2017*). A large study of Swedish scientists revealed that women are

**Table 1.** Summary of data on authors listed a contributing equally.

| Journal title | Article statistics | | | Contributed equally = 2 | | | | Contributed equally > 2 | | | |
|---|---|---|---|---|---|---|---|---|---|---|---|
| | Total | Unknown | Usable | mm | mf | ff | fm | m first | f first | all M | all f |
| Biophysical J | 101 | 2 | 99 | 60 | 17 | 5 | 9 | 5 | 0 | 3 | 0 |
| Cell Reports | 105 | 1 | 104 | 41 | 12 | 11 | 15 | 12 | 9 | 3 | 1 |
| Curr Biol | 103 | 4 | 99 | 41 | 23 | 14 | 12 | 2 | 4 | 2 | 1 |
| eLife | 90 | 2 | 88 | 25 | 18 | 15 | 11 | 5 | 6 | 8 | 0 |
| J Biol Chem | 300 | 28 | 272 | 99 | 52 | 35 | 48 | 15 | 11 | 6 | 6 |
| J Cell Bio | 101 | 3 | 98 | 22 | 35 | 14 | 17 | 4 | 4 | 2 | 0 |
| J Clin Invest | 121 | 6 | 115 | 42 | 19 | 11 | 19 | 12 | 6 | 6 | 0 |
| J Exp Med | 210 | 13 | 197 | 65 | 40 | 26 | 30 | 10 | 8 | 11 | 7 |
| J Immunol | 308 | 10 | 298 | 89 | 76 | 59 | 48 | 16 | 3 | 5 | 2 |
| mBio | 100 | 2 | 98 | 27 | 26 | 12 | 14 | 8 | 7 | 3 | 1 |
| Nature | 104 | 6 | 98 | 44 | 12 | 7 | 8 | 14 | 3 | 10 | 0 |
| PLOS Bio | 110 | 9 | 101 | 39 | 18 | 13 | 17 | 6 | 5 | 3 | 0 |
| PLOS Comp Bio | 95 | 2 | 93 | 45 | 14 | 6 | 11 | 3 | 4 | 10 | 0 |
| PLOS Genet | 186 | 8 | 178 | 52 | 26 | 16 | 23 | 20 | 35 | 6 | 0 |
| PLOS Negl Trop Dis | 105 | 6 | 99 | 32 | 13 | 21 | 19 | 8 | 4 | 2 | 0 |
| PLOS Pathogen | 179 | 7 | 172 | 35 | 39 | 33 | 25 | 20 | 9 | 7 | 4 |
| PNAS | 411 | 14 | 397 | 151 | 66 | 44 | 61 | 30 | 19 | 22 | 4 |
| Science | 128 | 11 | 117 | 34 | 15 | 21 | 25 | 7 | 9 | 6 | 0 |
| Initial search* | 57 | 0 | 57 | 0 | 35 | 0 | 22 | 0 | 0 | 0 | 0 |
| Misc† | 120 | 2 | 118 | 57 | 27 | 14 | 12 | 4 | 2 | 1 | 1 |
| | 3034 | 136 | 2898 | 1000 | 583 | 377 | 446 | 201 | 148 | 116 | 27 |

*These papers are from the early searches used to identify the variables in this study and only mf and fm numbers were recorded. These 57 papers were removed from subsequent analysis.

†Miscellaneous includes the following journals; the number of articles in which two or more authors made equal contributions is shown in parenthesis for each journal. American Journal of Pathology (1); Angewandte Chemie (17); Biochemical and Biophysical Research Communications (2); Blood (1); BMC Bioinformatics (1); BMC Proceedings (1); BMC Systems Biology (1); Brain Pathology (1); Cancer Research (2); Cell (6); EMBO Journal (4); European Journal of Immunology (1); FEBS Letters (27); Genes and Development (5); Genome Research (1); Hepatology (1); International Journal of Cancer (2); Journal of Bone and Mineral Research (2); Journal of Cell Science (2); Journal of Molecular Biology (1); Journal of Physiology (2); Memórias do Instituto Oswaldo Cruz (15); Nature Biotechnology (1); Nature Cell Biology (1); Nature Genetics (15); Nature Materials (1); Nature Medicine (4); PLOS One (1); Protein Science (1).

DOI: https://doi.org/10.7554/eLife.36399.002

The following source data is available for Table 1:

Source data 1. All data organized by journal, author/gender category, and year.
DOI: https://doi.org/10.7554/eLife.36399.003

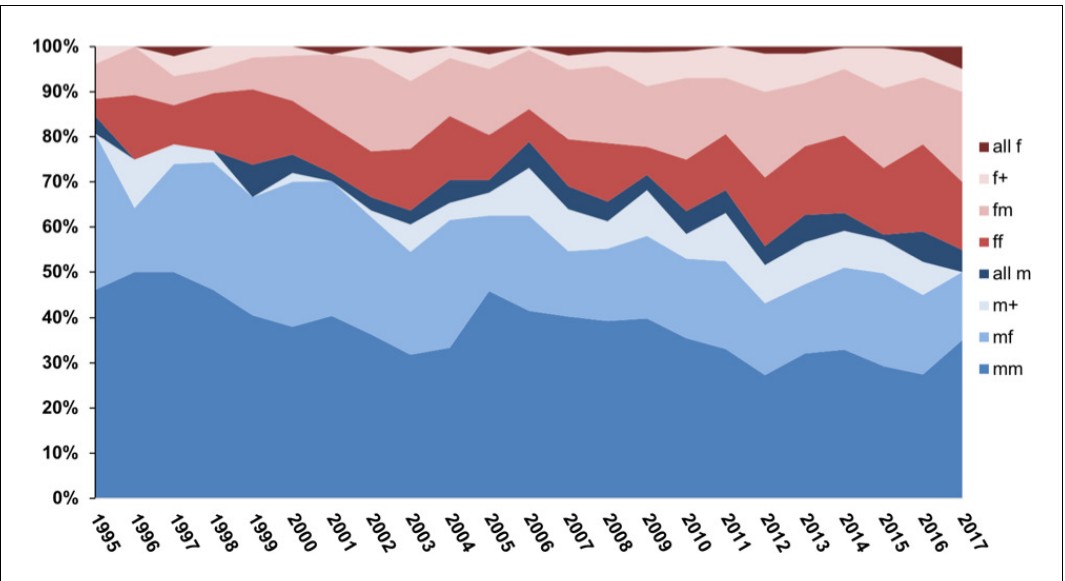

**Figure 1.** Proportion of various gender combinations among joint first authors in scientific papers published between 1995 and 2017. We studied papers in which two or more authors shared the first author position: 'fm', 'ff', 'mf' and 'mm' represent papers in which two authors shared the first author position, with the actual order of the authors being female-male, female-female, male-female and male-male. For papers in which more than two authors shared the first author position, 'all f' means that all these authors were female, 'all m' means they were all male, and 'f+' or 'm+' means that the first author listed in a mixed-gender combination was female or male respectively. The plot shows that the proportion of combinations in which a male author is listed first (various shades of blue) is decreasing over time.

DOI: https://doi.org/10.7554/eLife.36399.004

The following source data and figure supplements are available for figure 1:

**Source data 1.** Raw data for *Figure 1*.
DOI: https://doi.org/10.7554/eLife.36399.007
**Figure supplement 1.** Distribution of papers analyzed per year in this study.
DOI: https://doi.org/10.7554/eLife.36399.005
**Figure supplement 1—source data 1.** Raw data for *Figure 1—figure supplement 1*.
DOI: https://doi.org/10.7554/eLife.36399.006

more likely to be middle authors and less likely to be senior authors (*van den Besselaar and Sandström, 2017*). Hence, the available evidence suggests that disparities exist in gender contribution and position to the author byline of scientific publications.

In this study, we analyzed the gender order of publications where two or more individuals shared the first author position by stating that they had contributed equally. The expectation from equal contribution is that the order of author gender will be equally distributed or perhaps follow some ordering convention such as alphabetical order. Instead, we found a predominance of male authors at the first author position irrespective of whether first authorship was shared by two or more scientists. Furthermore, male-male author pairings and all male authors sharing equal credit was far more frequent than

corresponding female combinations. The finding of gender inequalities among authors who contributed equally suggests that inequities in credit sharing may be a contributing factor to the continuing gender imbalances reported for academic positions, grant funding, and awards. The results suggest a need for more clarity and transparency in stating how author position is selected when more than one author share equal credit.

## Results

We analyzed 3034 scientific publications from 1995 to 2017 where two or more authors stated to have contributed equally. From this set, we identified the gender for each of the authors listed as contributing equally in 2898 publications, which provided our usable dataset (*Table 1*). Two authors were listed as

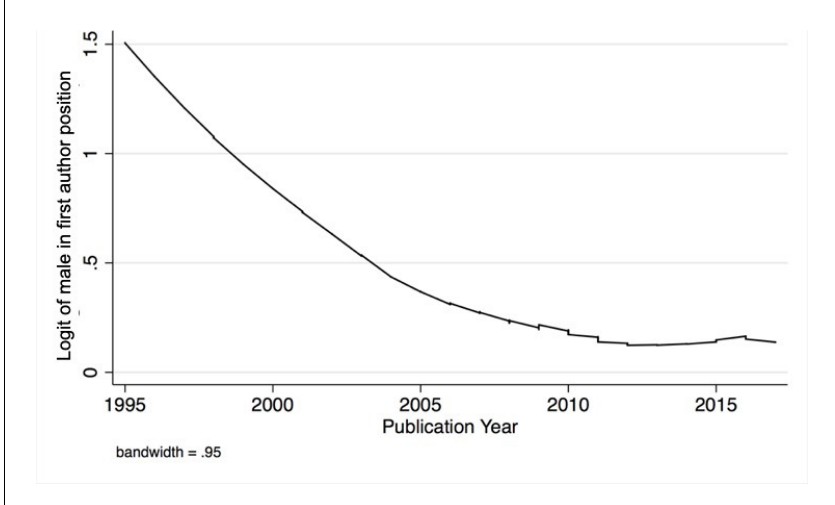

**Figure 2.** Gender bias in the first author position over time. Temporal trend in gender bias among two equally contributing authors of different gender: the y-axis is log (p/(1 p)) where p is the probability of bias; the x-axis is publication year. In the absence of bias, log (p/(1 p)) would be zero.

DOI: https://doi.org/10.7554/eLife.36399.008
The following source data is available for figure 2:

**Source data 1.** Data analysis tables used for *Figure 2*.
DOI: https://doi.org/10.7554/eLife.36399.009
**Source data 2.** Data analysis log.
DOI: https://doi.org/10.7554/eLife.36399.010
**Source data 3.** Data management file for data analysis.
DOI: https://doi.org/10.7554/eLife.36399.011

individuals of both genders contributed equally and that this author order was random, one would have expected roughly equal male-female and female-male author pairings. Comparing the expected and observed gender ratios yielded a Chi-square statistic of 15.8 (df = 1 and p<0.001).

Dividing the mixed gender two author publications into those published 1995–2006 and 2007–2017 yielded 190 and 358 pairings for male-female author order, respectively, and 103 and 321 pairings with female-male author order, respectively. When these ratios were analyzed with the Chi-square statistic the difference between observed and expected ratios was significant in the 1996–2006 group (p<0.001), but not in the 2007–2017 group (p=0.156). Analysis of publications for gender order using a Generalized Linear Population-Average model with Binomial Distribution and Logit-link estimated with Generalized Estimating Equations (GEE) and robust variance provided an estimate for gender bias in first authorship as the dependent variable among papers with two equally contributing authors of different gender (N = 972). The Odds Ratio of Gender Bias in First Authorship (95% Confidence Interval) using year as a continuous variable was 0.958 (0.931–0.986), indicating an estimated 4.2% decrease in odds of preference for males in the first position per year considering all publications from 1995 to 2017, adjusted for country, (95% CI: from 1% to 7% decrease per year, p-value=0.003). The Odds Ratio comparing publications after 2007 to those in years 1995–2007 (95% confidence interval) using the year as a categorical variable and adjusting for country was 0.605 (0.443–0.828) with p=0.002. Hence, the preference for males in the first position was pronounced prior to 2007, but has since decreased (*Figure 2*).

For the 492 publications with three or more authors contributing equally, the most common form involved mixed gender contributions (349, 71%). Of these 349 publications, 201 (57.59%) listed a male author first while 148 (42.40%) listed a female author first. Comparing male author first versus female author first ratios observed versus expected values yielded p=0.005. Although lower numbers precluded a decadal analysis these numbers imply a preference for male authors in the first positions of a multi-author byline when three or more individuals contribute equally. Analysis of the frequency of publications with three or more authors as a function of publication year revealed a positive trend line with time, suggesting that these author combinations are becoming more

contributing equally in 2406 (83%) publications, while 492 (17%) listed three or more (*Table 1*). We identified eight classes of author combinations claiming equal contribution: male-male, male-female, female-male, female-female, more than two all-male, more than two all-female and more than two with mixed gender having either a male or female listed first (*Figure 1*). For publications where two authors contributed equally, the most common gender pairing involved mixed gender, which was closely followed by male-male author pairings and female only author pairings were least frequent. These 2406 publications included 57 publications identified in an initial search to determine what could be expected in the literature, for which we only recorded gender order data. As such, they were removed from further data analysis, which gave us 2349 publications with two co-authors. Of these, 1377 papers had two authors of the same gender (mm or ff), leaving 972 with mixed gender authors claiming equal contribution, which consisted of 548 (56.38%) where the male author was listed first and 424 (43.62%) where the female author was listed first. Assuming that

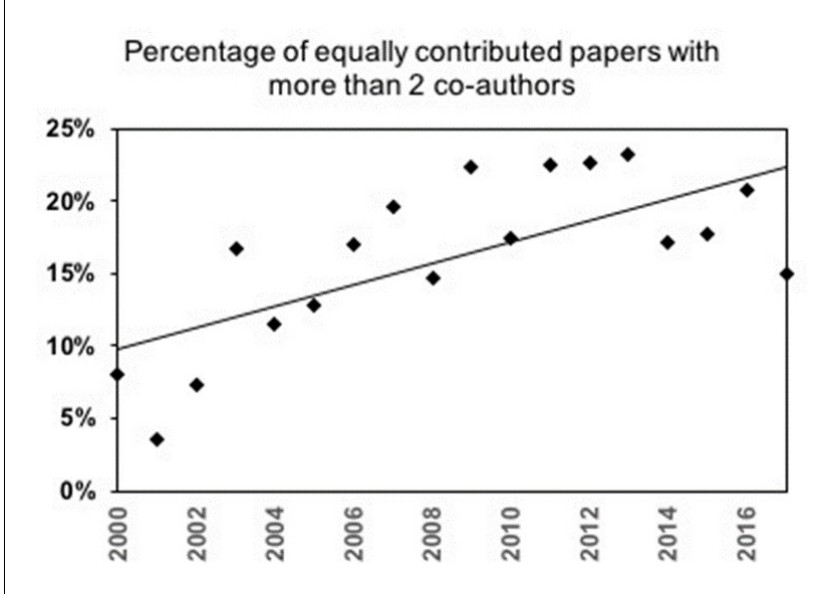

**Figure 3.** Percentage of papers with two or more authors contributing equally as a function of time. Points denote the percentage of papers with where more than two authors claimed equal contribution. Trendline $R^2$ value as 0.4857.

DOI: https://doi.org/10.7554/eLife.36399.012

The following source data is available for figure 3:

**Source data 1.** Raw data for *Figure 3*.
DOI: https://doi.org/10.7554/eLife.36399.013
**Source data 2.** Data analysis log.
DOI: https://doi.org/10.7554/eLife.36399.014
**Source data 3.** Data management file for data analysis.
DOI: https://doi.org/10.7554/eLife.36399.015

frequent (*Figure 3*). Analysis of likelihood of male preference in the first author positions when three or more individuals contributed equally did not reveal any statistically significant associations with year of publication or country. The results of the Generalized Linear Population-Average model described above showed a 1% decrease in odds of gender bias per year among publications with more than two equally contributing authors (estimated Odds Ratio = 0.99, 95% confidence interval: 0.933, 1.051).

Analysis of the relative distribution of the eight types of author association as a function of continent from which the publication originated revealed similar patterns for Asia, North America and Europe (*Figure 4*). Patterns for Africa, Europe, and South America were different from those of Asia, North American and Australia groupings, but some of these categories contain fewer papers, which suggests a need for caution in comparing between these continental groupings. We considered analyzing temporal trends for various countries, but with the exception of

the United States had too few per country for a meaningful analysis. Consequently, we divided the publications into three groups of origin, United States, Europe and Other. Analysis of the predicted probabilities of gender bias in first authorship by year and country among papers with two equally contributing authors of different gender showed a declining trend for each of the three world regions (*Figure 4*). In these analyses, there were no differences in gender bias by country.

We analyzed the frequency of alphabetical ordering for author sequences for all publications examined. Overall, 49.6% of all publications had the authors names ordered alphabetically. The percentages of author associations for which the author sequence was alphabetical were 49%, 49%, 48%, 55%, 22%, 41%, 26%, and 25% for male-male, male-female, female-male, female-female, three or more authors all male, three or more authors all female, three or more authors male author first, and three or more authors female author first, respectively. In comparing male vs, female author first position there was no significant difference between the frequencies of alphabetical versus no-alphabetical ordering. However, among those publications where the authors were ordered alphabetically male-female author combinations were more common than female-male author combinations (Chi-square statistic is 4.359; p=0.037).

## Discussion

Male-male author combinations were the most common gender combination and this applied to combinations involving both pairs and association of more than two authors. Female-female author combinations were the least common gender combination comprising less than half the number of observed male-male author combinations. Male-Female and Female-Male author combinations were almost as common as male-male author combinations, but the frequency differed in gender order. Male-female author combinations were significantly more frequent than female-male author combinations, with a ratio of 1.3:1. However, analysis of the data as a function of time revealed that the effect was strongest for publications dating before 2007. In the past decade, the preference for male gender in the first author position among mixed-gender pairings declined such that there was no statistical difference between observed gender order pairings and those expected from random

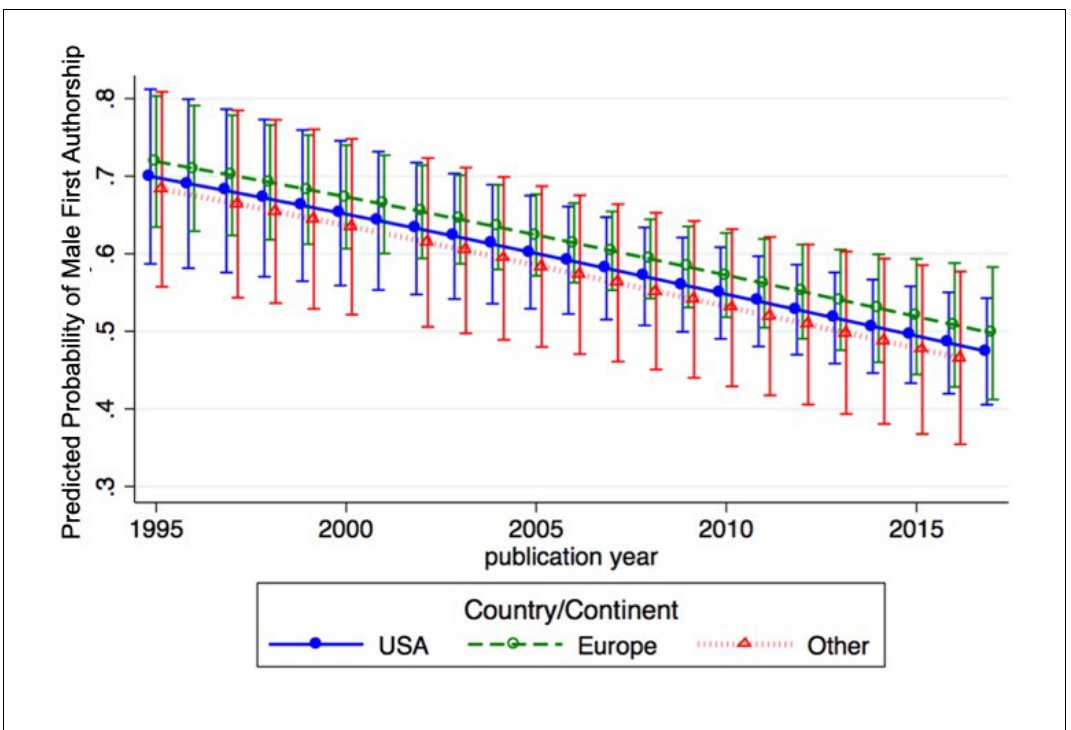

**Figure 4.** Gender bias in first authorship by year and geography. Predicted probabilities of male first authorship by year for three geographical regions (USA: blue; Europe: green; Other: red) among papers with two equally contributing authors of different gender (N = 972).

DOI: https://doi.org/10.7554/eLife.36399.016

The following source data is available for figure 4:

**Source data 1.** Data analysis tables used for *Figure 4*.
DOI: https://doi.org/10.7554/eLife.36399.017
**Source data 2.** Data analysis log.
DOI: https://doi.org/10.7554/eLife.36399.018
**Source data 3.** Data management file for data analysis.
DOI: https://doi.org/10.7554/eLife.36399.019

assignments. Whereas pairings of two authors sharing equal contribution for the first author position comprised the majority of combinations, we found a significant minority for author combinations involving three or more authors contributing equally. As with single pairings, the preference for males in the first author position also occurred with author combinations of three or more authors listed as contributing equally, with a ratio of 1.35:1. Male only combinations were also more common when three or more authors contributed equally such that there were almost four times more male only combinations than female only combinations.

Although we have no information on the mechanism for selection of these gender author assignments the disequilibrium between observed and expected ratios strongly suggests that these selections were not made randomly.

In fact, only one of the publications we analyzed provided the rationale for the author order and indicated it was based on alphabetical ordering (see, for example, *Bieberstein et al., 2012*). In the biological sciences, the first author position is the person who is the most important contributor to the study and these are often students and postdoctoral fellows who are considered trainees. Hence, having first author publications is measure of success during the graduate and postdoctoral training years that can affect career trajectories. It is conceivable that some of the disequilibrium in gender ratio in the earlier years of this study reflects an inherent bias towards males that was a residual result of seniority differences at the time that the work was prepared for publication, as males have comprised a higher percentage of trainees until recent decades (*National Research Council, 2000*).

However, such an explanation would not apply for most papers published in the 21st century as the proportion of women trainees in the biological sciences, which this data set reflects, has exceeded that of men since 2000 (*Committee on Women in Science, Engineering, and Medicine, 2019*).

Given the importance of the first author position in credit allocation for publications in the biomedical sciences, the disparity in frequency between male-female and female-male author combinations raises the possibility for unequal gender benefit among combinations sharing the first authorship despite these being designated as contributing equally. We note that information on equal contributions is often included as a footnote and for some publications it is stated only in the pdf and print versions and are thus absent from the online full text versions, which are increasingly the format readers access. Consequently, it is likely to that the author listed as contributing equally in the second position may be not benefit as much as the author listed first, which is usually recognized as the most important contributor in biomedical publications. Recognizing this issue, some journals are encouraging citations to indicate both authors in equally contributed publications, instead of the traditional 'firstname *et al.*' format (*Drubin, 2014*).

The finding of a disproportionate number of males in the first author position relative to expected numbers had these positions been selected randomly is consistent with several studies showing that female authors receive less credit recognition relative to their male author colleagues. A study of PhD students revealed that male graduate students were 15% more likely to be listed in publications than their female counterparts (*Feldon et al., 2017*). An analysis of male and female authorship patterns for publications in natural sciences, social sciences, and the humanities showed that a large predominance of male author over female authors in the first and last positions (*West et al., 2013*). Perhaps most relevant for our findings is the observation that women receive less credit than men for team work in academia (*Sarsons, 2017*). The finding that the preference for male first publications had declined in the past decade could reflect gains by women in academia in recent years. Nevertheless, given that authorship position in a scientific paper can have career altering consequences, choices made years ago could have long lasting effects that may still be a

contributing factor to current gender inequities in academia.

We observed that male-only author pairings were more common in author combinations of two or more authors. This finding cannot be explained by larger numbers of male trainees, as women have exceeded men in PhD training since 2000 (*Committee on Women in Science, Engineering, and Medicine, 2019*). Again, without access to how these orderings were decided, or to the gender composition in the laboratories, we cannot infer the causes for this gender preference. Nevertheless this observation is consistent with the finding that males are more likely to share data with other men, which can lead to scientific discussions and collaborations that result in shared first author publications (*Massen et al., 2017*). The high prevalence of publications sharing first authorship among three or more males echoes the concern that male-exclusive networks exist in science (*Massen et al., 2017*).

The frequency of multi-author equal contributions dropped rapidly for combinations of more than three authors, but we observed at least two groupings of 11 authors (*Dastani et al., 2012*; *Ohlsson et al., 2011*). We noted a positive trend for the frequency of publications listing three or more authors contributing equally, suggesting that such author combinations may be increasing as a function of increased team science in biomedical research. We note that some have questioned whether statements of equal author contribution can ever be accurate given the problem of weighing the relative value of different contributions (*Moustafa, 2016*). A recent analysis of journal instructions for authors revealed that none addressed equal contribution statements (*Resnik et al., 2016*). Our findings of a disequilibrium between observed and expected male and female authorship positions among groups of authors that contributed equally suggests a need for explicit requirements that explain how the ordering is done.

The majority of publications analyzed came from the United States, reflecting the predominance of this country in contributing to the biomedical literature. Analysis of distribution of author combinations for the continent of origin produced similar patterns for Asia, Australia and North America, which may reflect similar practices in author order selection in the contributing countries. We note with interest that the patterns for Africa, Europe, and South America differed from the Asia, Australia, and North American groupings, but caution against

drawing conclusions since for some of these continent groupings the number of publications analyzed may not be adequate to make direct comparisons. Nevertheless, the possibility that there are differences in author gender order and combinations depending on country of publication is an interesting area for future investigation.

We acknowledge some limitations in our study, which suggest caution in interpreting the data. The finding that the preferences for males and females in the first author position varied over time, suggests that variables contributing to these decisions may be changing rapidly. We noted differences between journals in the proportion of pairings, suggesting that that there may be differences between fields that could skew results depending on the source database. Limitations on the use of search engines are discussed in the Materials and Methods. Our findings should be complemented by subsequent studies, which may be able to analyze a larger number of publications across many disciplines through automated searches linked to gender image recognition software. In this regard, a recent study that examined papers published between 2005 and 2014 with co-first authors of different genders using online databases in addition to our method of website images, found that there was no difference in female versus male authors in the first position in basic science journals, which is consistent with our results, but that female co-first authors were less represented in the first position in clinical journals (*Aakhus et al., 2018*). As such, our finding that a disequilibrium exists in gender ratios among authors listed as contributing equally in the first position is sufficiently robust to raise concern on the fairness of shared credit contributions assignments in biomedical publications. At the very least, this study opens a window into a relatively unexplored area in the sociology of science that could have major consequences for current efforts to improve gender equity in academia.

In summary, our results provide evidence that the first position of author bylines involving mixed gender combinations contributing equally to a publication is more likely to have a male author. We note that the disequilibrium in gender ratios among authors who contributed equally has abated in the past decade. This is certainly good news and suggests that the problem may be going away. However, milestones in scientific careers such as hiring and promotion often occur many years after publications in

graduate school and postdoctoral training when scientists are likely to be in the first authorship position. Consequently, it is possible that the effects of the disequilibrium measured in this study will linger for some time to come with disproportionately negative effects on women who shared first authorship and appeared second in the author byline. Given the importance of first authorship in biomedical publications and the increasing popularity of sharing authorship with the rise of team science a male preference could have consequences on hiring decisions, promotion and the distribution of resources such as grant funding. This information should be of interest to promotion and grant review committees as they consider the merit of applications who list papers stating that they contributed equally. The finding of gender inequalities among authors who contributed equally raises the possibility that some authorship decisions are vulnerable to conscious or unconscious biases and this suggests the need for journals to require statements of how author ordering was done in publications claiming equal contributions.

## Materials and methods

The study was done in three stages. First, we undertook a cursory review of publications using the Google Scholar search engine with the keywords 'contributed equally' to familiarize ourselves with the variables involved and get a sense as to whether there were differences in how often males and females shared first author positions. This stage involved analyzing several hundred publications, which identified 57 publications that had one or more co-first authors (listed in *Table 1* as results from 'early searches'). This initial analysis revealed that whereas our initial interest was in gender positions among mixed gender pairs contributing equally there were many publications with more than two authors, suggesting the need for analyzing different journals. These 57 publications were not used in the statistical analysis. In the second stage, we undertook a search of papers using two search strategies. One strategy used Google Scholar to search for the keywords 'contributed equally' and a specific journal name.

The second strategy searched for the phrase 'contributed equally' in individual journal websites. After finding several hundred publications, we compared the results of the two search strategies and found discrepancies. Specifically, the Google Scholar search strategy was returning a

higher frequency of male-female (M:F) orderings among those chairing first position than the in journal website search strategy. Inspection of the identified articles revealed that that the Google Scholar search strategy was returning more older papers suggesting a temporal variable to male-female author orderings, a finding that was subsequently confirmed at the conclusion of the study.

The third stage of the study involved adding more papers using both the Google Scholar search and journal website strategies with the searches targeting specific years for those years where few papers were initially identified. The journals selected in this study were chosen because they are widely known in the biomedical sciences: criteria for a journal being well known included sponsorship by a scientific society, a long history of publication and/or a significant journal impact factor (*Supplementary file 1*).

We note some caveats in the approach to search citations, which can affect the results depending on the method used. Using search engines such as Google Scholar facilitates searches since searching for the words 'contributed equally' in journal sites usually identifies many irrelevant publications where these words are in the text of the article. However, using the search engine introduces potential biases depending on how the algorithm prioritizes those publications containing the words 'contributed equally'. Many of the results from those searches were not usable in this study because they related to shared internal and corresponding author contributions and to the use of the search phrase in the text of the paper. For some publications, authors were listed using first initials and it was not possible to assign gender. We could not reliably assign gender to authors from name alone in approximately 4% of publications. We note that the percentage of gender non-identification in our study compares favorably with the 5, 6 and 17% uncertainties reported in other studies using different methods (*van den Besselaar and Sandström, 2017*; *West et al., 2013*; *Bendels et al., 2018*).

One of the two coauthors inspected each article manually. Determination of author gender was done by searching for images of an individual's name using the Google search engine, which was adequate to assign gender for 97% of papers examined. Searches were narrowed by including the name of the research institution or research subject among the search words. In many instances, we were able to locate the individual by finding the website for the laboratory producing the paper. For those individuals whose gender could not be identified the major cause for failing identification was the absence of a photograph on the web. For each paper, we recorded the country of origin based on the country of the corresponding author, the gender of individuals sharing the first authorship, the year of publication, and whether the order of authors sharing equal credit was alphabetical. We estimate that the analysis of each entry averaged approximately 5 min, since each publication needed manual inspection to confirm that the search engine was correct and this often necessitated inspecting the PDF version of the publication, as author contributor information was not often available for the online format versions.

At the beginning of this study, the Pubmed database had approximately 25,000,000 million entries. However, given that only 0.8% of all papers in PubMed have first authors that contribute equally (*Dubnansky and Omary, 2012*), this would reduce the size of the database of interest to approximately 200,000 publications. Hence, our database of 2898 usable publications represents about 1.45% of the available publications in this category. Chi-square analysis used the on-line calculator http://www.socscistatistics.com/tests/chisquare/Default2.aspx

### Statistical method

Gender bias in first authorship was defined as first male author when all the authors contributed equally. Generalized linear population-average model with binomial distribution and logit link was used to estimate odds ratio of gender bias associated with country of origin and year of publication. The model was estimated using generalized estimating equations (GEE) and robust variance estimates. The analysis accounted for potential correlation among multiple publications by the same journal. The analysis was stratified by the number of authors: two versus more than two. Country of origin was modeled as a categorical variable with two indicator variables in the model and US as the reference category: Europe vs. US and Other vs. US. The publications that included multiple countries, such as European and the US, were classified as US-based. Year of Publication was modeled as a continuous linear term in the model as well as a two-category predictor: 2007–2017 vs 1995–2006. Analysis was performed using STATA version 15, StataCorp.

2017. Stata Statistical Software: Release 15. College Station, Texas: StataCorp LLC.

## Acknowledgements
We are very grateful to Ms Abanti Sanyal and Dr Gayane Yenokyan of the Johns Hopkins Biostatistics Center for statistical analysis and advice on revising this paper.

**Nichole A Broderick** is in the Department of Molecular & Cell Biology, University of Connecticut, Storrs, United States

nichole.broderick@uconn.edu

http://orcid.org/0000-0002-6830-9456

**Arturo Casadevall** is in the Department of Molecular Microbiology and Immunology, Johns Hopkins School of Public Health, Baltimore, United States

acasade1@jhu.edu

http://orcid.org/0000-0002-9402-9167

*Author contributions:* Nichole A Broderick, Arturo Casadevall, Conceptualization, Data curation, Formal analysis, Investigation, Methodology, Writing—original draft, Writing—review and editing

*Competing interests:* The authors declare that no competing interests exist.

### Funding

| Funder | Author |
| --- | --- |
| Johns Hopkins Bloomberg School of Public Health | Arturo Casadevall |
| University of Connecticut | Nichole A Broderick |

The funders had no role in study design, data collection and interpretation, or the decision to submit the work for publication.

### Decision letter and Author response

Decision letter https://doi.org/10.7554/eLife.36399.024
Author response https://doi.org/10.7554/eLife.36399.025

## Additional files

### Supplementary files

• Supplementary file 1. Characteristics of journals chosen for this study.
DOI: https://doi.org/10.7554/eLife.36399.020

• Transparent reporting form
DOI: https://doi.org/10.7554/eLife.36399.021

## Data availability
Source data files for all figures have been provided along with the complete raw dataset.

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
