## [Decision Letter]

Thank you for submitting your article "Disequilibrium in Gender Ratios among Authors who Contributed Equally" to *eLife* for consideration as a Feature Article. Your article has been reviewed by three peer reviewers, and the following individual involved in review of your submission has agreed to reveal her identity: Sandra Masur (Reviewer #1).

We would like to invite you to submit a revised manuscript that addresses the concerns raised by the reviewers – please see below.

Summary:

The authors assembled a data set of 3035 articles published between 1995 and 2017 in a variety of biomedical journals in which 2 or more authors were indicated as having contributed equally. They assigned genders to authors manually by finding online photos of authors for 97% of papers and compared rates of male first authors versus female first authors in a series of analyses. The authors are to be commended for tackling an important topic. There are, however, concerns about aspects of the study designs and the techniques used to analyse the data (see below). The manuscript also contains a surprising number of simple errors.

Essential revisions:

Study design and methods:

1) The authors refined and changed their hypotheses and analytical plan while examining data. This would be fine if they had treated the preliminary data as a pilot study from which they designed their study. However, they appear to have included the original data in their full data set.

2) The methods for assembling the data set are not well described, and from the available description, they seem far from systematic. Phases 1 and 3 are particularly problematic, especially Phase 3, as this came after the majority of data collection. The authors claim their search methods for Phase 2 were systematic but provide few details of the system used. Such necessary details would include: a) On what basis were journals selected? (The authors note that journals were selected because, "they are widely known in the biomedical sciences." Can they support this statement in any way?)b) Did the authors include all google scholar and journal website search results or only a certain number of articles or pages? c) How did they select included papers? Was this done by a single person (not recommended) or did they use two independent analysts (recommended)? If the latter, what was their kappa score?

The authors may wish to consult standard methods of a systematic review to gain a better understanding of what is typically expected for a search to qualify as systematic. (E.g., see: Cochrane Handbook Chapter 6.4, which is freely available online.)

Inferential statistics:

3) A number of the analyses in the manuscript are, unfortunately, unacceptable. I would suggest that the authors consult with a statistician or epidemiologist to help them consider their analytical options, including relatively simple methods such as logistic regression, or perhaps something like generalized estimating equations, to account for any instances of the same author(s) appearing multiple times. With such analyses, they could then include year of publication in the model. This would allow them to draw much more robust findings and conclusions that would strengthen their paper and the impact of all the work they have clearly put into this manuscript. (In my experience, many researchers in the biomedical sciences are not aware of the statistical services that are available at their own institution, so I include the following links in case that is the case here:https://www.jhsph.edu/research/centers-and-institutes/johns-hopkins-biostatistics-center/services/index.html;https://stat.uconn.edu/consulting-info/)

4) The authors have conducted multiple Chi-squared analyses without accounting for the ensuing inflated potential for Type I error. Also, they do not report the Chi-squared statistic and degrees of freedom, only the p-value.

5) The analyses do not account for overall gender composition of the pool of authors. If we only examine the 2-author papers and assume that each author is unique, there are 2*1000 + 581 + 447 = 3028 male authors compared to 2*377 + 581 + 447 = 1782 female authors. (Please also consult work by Sugimoto and Larivière for systematic analyses of male and female authorship patterns.) Is it really therefore surprising that more papers are led by male authors? The authors refer to the proportion of women trainees (citing reference 19) but provide no citation to support their implication that first authors may reasonably be assumed to be exclusively trainees.

6) The analyses do not account for the same person or people authoring multiple papers. Is this possible in this data set? If yes, this potential clustering effect could have been accounted for had they used a multilevel model for their analyses. See also point 3 above.

7) Changes in authorship credit allocation over time are accounted for in a very crude way by splitting the data set into two sets, dividing at 2007. This is not an appropriate way to account for change over time, and only adds to the problem of multiple hypothesis testing. See also point 3 above.

8) The authors have split the dataset into articles with 2 equally-contributing authors and those with more than 2. Like the time-based split, this is problematic. Do the authors have any basis for assuming that there is a difference between these types of credit-sharing such that they must be analyzed separately?

9) In the Results section, the authors note that they looked at the possibility for alphabetical order in a subsample of 2109 papers. Why this subsample? How was this selected?

10) The authors report a confidence interval in their methods but do not apply it in any way in their presentation of results. More worrisome is that the authors have not accounted for the fact that confidence intervals rely on the assumption that the sample population (i.e., the data analyzed) was sampled randomly from the population. This was emphatically not the case in this study. For this reason, the confidence interval calculation does not apply. (A minor point in relation: I was able to reproduce their estimate of a confidence interval of 2.38 using the website they used by leaving the default setting of 50%, which was not the result in their study. This suggests a potential lack of understanding of how to use these calculations.)

Descriptive statistics:

11) Four out of six continents lack sufficient sample size for it to be valid to look at distribution of shared authorship even descriptively. In addition, presenting these descriptive statistics in equally-sized pie charts inadvertently conveys that each continent has an equivalently-sized subsample in the data set.

Conclusions:

12) The data presented in this paper suggest that any gender bias in author order (which certainly appears to have been a legitimate issue in papers published 1995-2007) is no longer a problem in more recent papers. If this is true, this is excellent news and should be presented as such.

[Editors' note: further revisions were requested prior to acceptance, as described below.]

Thank you for submitting the revised version of your manuscript "Disequilibrium in Gender Ratios among Authors who Contributed Equally". The revised manuscript has been reviewed again by the referee who had the most specific concerns about the statistical analyses employed in the original version (Reviewer #2), and while they welcome many of the changes, they still have a number of concerns that you will need to address (please see below). There are also several editorial points I would like you to address (again, please see below). Although this might seem like a long list (24 points), most of them should be straightforward to address.

Reviewer #2:

The authors have nicely addressed some of my concerns (in particular the statistical analyses are improved-the authors are to be commended for their work to improve the robustness of their conclusions) but some concerns remain, and some minor new concerns arose pertaining to the new analyses. My main concerns remaining have to do with how the statistical analyses are presented.

1) First, for their new analyses, the authors have named their primary outcome variable "gender bias" and defined it as follows: "gender bias in first authorship was defined as first male author when all the authors contributed equally." This is problematic and does not align with how gender bias is defined in other literature. In an ideal world, there would be no gender bias, meaning that the optimal value of a variable named "gender bias" would be zero. Yet, presumably, the authors are not arguing that no man should ever be listed first under conditions of shared first authorship. By naming this variable "gender bias," this is what is currently implied. It would be more understandable to readers (and more in keeping with how this is usually done in studies of sex and gender in humans) to define this variable as something like, 'male author listed first' or perhaps 'm* authorship' in keeping with their notation used elsewhere in the paper. This makes it more intuitive to the reader that the ideal level for this variable is 0.5, not 0.0.

2) Second, it is standard practice to provide a table in the results showing odds ratios and confidence intervals (along with p-values, if one believes those are important, which the authors seem to) for each variable in the model. This would help readers who are familiar with these kinds of statistics more rapidly grasp the findings.

I list these two points first and second because being clearer about how the outcome variable is defined and the odds ratios for each variable in the model could help solve some the following problem.

3) In the Results section, the authors state, "The Odds Ratio of Gender Bias in First Authorship (95% Confidence Interval) using year as a continuous variable was 0.958 (0.931-0.986) with p < 0.01, indicating a significant preference for males in the first position considering all publications from 1995-2017." The statistics presented do not support the authors' statement here. Odds ratios under 1 mean that the outcome is less likely than the reference, not more likely. In other words, the odds ratio presented in this sentence indicates that the variable "gender bias" is less likely than the reference. The authors may want to check with their statistical consultants on this, because based on the raw data presented a few lines up (56% male author first vs. 44% female author first in the n=971 mixed-gender subsample) I would not be surprised to see an odds ratio and 95% confidence interval of 0.958 (0.931-0.986) for female first authorship in that subsample.

4) Re: the alphabetical subset. The authors have added in the manuscript, "This number differs from the larger data set used in the analysis because we began to record alphabetical ordering after the study had begun when we realized this could be an important variable." I appreciate that they have explained why this happened. However, for this to be a complete, robust study, why don't they just quickly go back and record alphabetical or non-alphabetical ordering for the relatively small proportion of early papers (about one quarter of their sample? The fact that they didn't do it at the time doesn't preclude them from doing it now. This is particularly important given the journal's requirement to support replicability. Nowhere in the materials is it clear which papers in the dataset are part of the 2109 for which alphabetical or non-alphabetical order was recorded and which are part of the 787 for which such data were not collected at first glance.

5) More details are needed regarding the data and analyses to ensure replicability. What statistical software (including version) was used? Typically, when writing about statistical analyses, we include the name of the software (e.g., SAS, Stata, SPSS, etc.) or statistical programming environment (R) and version number. This is because the same analyses run in different software could produce slightly different results.

6) Related to the above, is it possible to provide the script or code as an appendix along with the data? This is good practice for ensuring replicability.

7) Results section: The authors state, "Comparing the expected and observed gender ratios yielded a Chi-square statistic of p <0.00.1" A p-value is not a Chi-squared statistic. A Chi-squared statistic would look something like Chi-squared(degrees of freedom)=number like 2.7 or 0.3 or 34.2 or 976.

8) Results section: "estimated 4%" should say "estimated 4% per year" and "1 to 7% decrease" should read "1 to 7% decrease per year."

9) The source data for Figure 2 doesn't appear to match the Figure. The Figure looks like it was plotted using year of publication as a quasi-continuous variable (i.e., there's one point for each year in the data set) while the source data shows year as a categorized variable (pre-2007 and 2007+) along with country/group of countries as a categorized variable.

10) Point 11 noted, "Four out of six continents lack sufficient sample size for it to be valid to look at distribution of shared authorship even descriptively. In addition, presenting these descriptive statistics in equally-sized pie charts inadvertently conveys that each continent has an equivalently-sized subsample in the data set." The authors responded, "We agree with the reviewer's comment. Therefore, in our new analyses, we looked at the country as a three-category predictor: Europe, US and other. The results for these regions matches are comparable suggesting that the trends we are observing are occurring worldwide." The analysis in the revision has addressed this very well and we now have a very nice Figure 5. However, Figure 4 continues to present 6 continents. This means that Figure 4 continues to have the problem of inadvertently conveying equivalency, and, in addition, it now fails to reflect the analyses conducted. I recommend simply removing Figure 4.

11) The original review stated: "16) If grammatically possible, please refer to 'women' or 'female authors' rather than 'females', and 'men' or 'male authors' rather than 'males.'" The authors' response to this was: "Response: Agree. Done." Yet, there are 22 instances in the manuscript of using 'males' or 'females' when it would be completely grammatically possible to refer to 'male authors,' 'men,' 'female authors,' or 'women.' For example, the second sentence in the revised abstract begins, "For mixed gender pairs males were…" A correct rewording of this line would be, "For mixed-gender pairs, male authors were…" It is acceptable to refer to mice as 'males' and 'females' but it is not acceptable to refer to humans in this way in most English-speaking countries, nor in scientific literature describing studies that involve humans as subjects, objects, or participants (e.g., in clinical medicine, social sciences, etc.). Please correct all 22 uses of 'males' and 'females'.

12) Point 16 in the original review noted that in the original Introduction, the authors stated that, "Analysis of articles in 5 medical journals showed that whereas papers listing equal contributions comprised less than 1% of publications in 2000, by 2009 this trend had increased to 3.6-8.6% depending on the journal (8)." In fact, the paper referenced by Akhabue and Lautenbach reports rates from 1.0-8.6% in 2009 in the top 5 general medicine journals. The 1.0% rate is from the BMJ. This means that the rate of co-first authorship in 2009 was 1.0-8.6%, not 3.6-8.6% as claimed by the authors. The change that they made in their revision (removing the reference to the 5 top general medicine journals) has not fixed the problem that their paper makes a claim unsupported by the citation they are using. They need to change 3.6% to 1.0% in order to be accurate.

13) In the file with file name Table 1–source data 1 what does the grey colouring on some rows mean? This should be specified so that anyone aiming to build on or replicate this work can understand the data file.

Editorial points to address:

14) Please consider changing the title to the following:

“Gender Inequalities among Authors who Contributed Equally”

15) Please consider revising the abstract to read as follows:

“We analyzed 2898 scientific papers published in the period 1995-2017 in which two or more authors shared the first author position. For papers in which the first and second authors made equal contributions, mixed gender combinations were most frequent, followed by male-male and then female-female combinations. For the mixed gender combinations, there were more male authors than female authors in the first position, although the disparity was less in the second decade of the period studied. For papers in which three or more authors made equal contributions, there were more male authors than female authors in the first position, and more all-male than all-female combinations. We also show that the disequilibrium in gender ratios among authors who made equal contributions is not consistent with random or alphabetical ordering of these authors. These results raise concerns about female authors not receiving their fair share of credit for scientific papers, and suggest a need for journals to request clarity on the method used to decide author order among those who made equal contributions.”

16a) Please consider replacing the term "disequilibrium in gender ratios" with the term "gender inequalities".

16b) Please consider replacing the terms associations, author associations etc. with the terms combinations, author combinations.

17) Please reword the phrase "the person who does the actual work".

18) Re the sentence that starts: "It is conceivable that some of the disequilibrium in gender ratio in the earlier years…", is it worth mentioning here or somewhere else in the manuscript that, irrespective of the gender breakdown of the science workforce, when considering just those manuscripts with two first authors, one would expect mf to equal fm?

19) Please reword the following sentence to be more precise and/or avoid the word "predominated" (and similar words): "We observed that male-only pairings predominated in author combinations of two or more authors."

20) The sentence "The frequency of multi-author equal contributions dropped rapidly for associations of more than three authors but we observed at least two groupings of 11 authors [Bonham and Stefan, 2017; van den Besselaar and Sandstrom, 2017]." suggest that Bonham and Stefan, 2017 and van den Besselaar and Sandstrom, 2017 have 11 authors, but this is not the case: please clarify.

21) Please move the passage about the limitations caused by the use of Google Scholar ("The approach to search […] in the text of the paper.") to the Materials and methods section, and add a short sentence to the main text stating that this limitation is discussed in the Materials and methods section.

22) Please expand the caption for Figure 2 to better explain to the reader what is shown in this Figure. The revised caption should include a title sentence (in bold). Also, please explain in the caption what the ideal level of this logit function would be. Please also define p(bias).

23) Please expand the caption for Figure 3 to better explain to the reader what is shown in this Figure. The revised caption should include a title sentence (in bold).

24) Please expand the caption for Figure 4 to better explain to the reader what is shown in this Figure. The revised caption should include a title sentence (in bold). Please also explain how these values are predicted.

[Editors' note: further revisions were requested prior to acceptance, as described below.]

I am pleased to inform you that your article, "Gender Inequalities among Authors who Contributed Equally", has been accepted for publication in *eLife*, subject to addressing a small number of points raised by the referee.

*Reviewer #2:*

Thank you very much for the opportunity to re-review this paper. The authors have responded extremely well to nearly all my comments, and I believe the paper is substantially improved. There are just a few small corrections remaining:

1. Regarding this passage in the Discussion: “We noted differences between journals in the proportion of pairings…” […]. It's up to the authors, of course, but I think they are underselling their results at this point by calling their findings "preliminary". Also, software for gender identification often uses names as well as, or instead of, images. I would suggest changing, "Hence, our findings should be considered preliminary until confirmed by subsequent studies, which may be able to analyze a larger number of publications across many disciplines through automated searches linked to gender image recognition software," to something like, "Our findings should be complemented by subsequent studies, which may be able to analyze a larger number of publications across many disciplines through automated searches linked to gender recognition software."

2. In discussing their findings in the context of the broader literature, the authors may wish to refer to a similar study published in JAMA, in February 2018, that came across my radar recently: https://www.ncbi.nlm.nih.gov/pmc/articles/PMC5838607/

---

## [Author Response]

Essential revisions:1) The authors refined and changed their hypotheses and analytical plan while examining data. This would be fine if they had treated the preliminary data as a pilot study from which they designed their study. However, they appear to have included the original data in their full data set.

In our new analysis, the data were treated as the reviewer suggested. We removed the preliminary data from analysis (indicated as ‘Initial Search’ in Table 1). We have adjusted the text to reflect how data were handled. The data from the pilot analysis are included in the table, but both figure/table legends and text indicate that they were removed for subsequent statistical testing. The text has been written more clearly in the methods and results to indicate how the data were treated. The removal of those early papers had no effect on the conclusions of the study.

2) The methods for assembling the data set are not well described, and from the available description, they seem far from systematic. Phases 1 and 3 are particularly problematic, especially Phase 3, as this came after the majority of data collection. The authors claim their search methods for Phase 2 were systematic but provide few details of the system used. Such necessary details would include:a) On what basis were journals selected? (The authors note that journals were selected because, "they are widely known in the biomedical sciences." Can they support this statement in any way?)

These journals were selected because they are prominent in the biomedical sciences and are well known. Many are society journals. To address this point, we have provided impact factor, society sponsoring and date of first publication in Supplemental file 1. In addition, we have added a sentence to the methods explaining why these journals were chosen.

b) Did the authors include all google scholar and journal website search results or only a certain number of articles or pages?

Generally, we limited ourselves to a set number of publications – usually 100 although in some cases we did more. In particular, we went back to include articles from dates ranges that were underrepresented in our initial search to provide adequate representation across time.

c) How did they select included papers? Was this done by a single person (not recommended) or did they use two independent analysts (recommended)? If the latter, what was their kappa score?

All papers discovered were included. Both authors worked to generate the database but analyzed different journals. We cannot calculate a kappa score because both authors took on different journals.

The authors may wish to consult standard methods of a systematic review to gain a better understanding of what is typically expected for a search to qualify as systematic. (E.g., see: Cochrane Handbook Chapter 6.4, which is freely available online.)

We have reviewed the suggested approach in Cochrane Chapter 6.4 (http://handbook-5-1.cochrane.org/chapter_6/6_4_designing_search_strategies.htm). Our study includes many of the suggested steps including random selection of articles etc. However, we agree that when we began the study, we did not set out to do the work with all the recommendations. We feel that our methods remain valid. We have removed the word ‘systematic’ from the text.

3) A number of the analyses in the manuscript are, unfortunately, unacceptable. I would suggest that the authors consult with a statistician or epidemiologist to help them consider their analytical options, including relatively simple methods such as logistic regression, or perhaps something like generalized estimating equations, to account for any instances of the same author(s) appearing multiple times. With such analyses, they could then include year of publication in the model. This would allow them to draw much more robust findings and conclusions that would strengthen their paper and the impact of all the work they have clearly put into this manuscript. (In my experience, many researchers in the biomedical sciences are not aware of the statistical services that are available at their own institution, so I include the following links in case that is the case here:https://www.jhsph.edu/research/centers-and-institutes/johns-hopkins-biostatistics-center/services/index.html;https://stat.uconn.edu/consulting-info/)

We consulted biostatisticians affiliated with the Johns Hopkins Biostatistics Center and they conducted independent statistical analysis, including fitting statistical models to the data. In particular, generalized linear population-average models with binomial distribution and logit link were used to estimate odds ratios of gender bias in first authorship associated with country of origin and year of publication. The model was estimated using generalized estimating equations (GEE) and robust variance estimates. The analysis accounted for potential correlation of multiple publications within the samejournal. The analysis was stratified by the number of authors: two versus more than two. Country was modelled as a categorical variable with 2 indicator variables in the model and US as the reference category: Europe vs. US and Other vs. US. The publications that included multiple countries, such as European and the US, were classified as US-based. Year of Publication was modelled as a continuous linear term in the model as well as a two-category predictor: 2007-2017 vs 1995-2006, based on exploratory analysis of the data. The statistical method section of the paper has been updated accordingly.

4) The authors have conducted multiple Chi-squared analyses without accounting for the ensuing inflated potential for Type I error. Also, they do not report the Chi-squared statistic and degrees of freedom, only the p-value.

The analytic approach has been updated. We primarily present the results of generalized linear population-average models with GEE that show estimated odds ratios of gender-bias associated with the primary predictors: year of publication and country, after accounting for potential clustering by journal. We also show two chi-square goodness-of-fit tests: one for two-author and another for more than two-author publications. The results are as follows: for two-author publications, chi-square statistic is 15.1 (with 1 degree of freedom) and p-value < 0.001. For more than two-author publications, chi-square statistic is 7.77 (with 1 degree of freedom) and p-value = 0.005. We did not adjust for multiple testing due to preliminary nature of these analyses. However, even if highly conservative Bonferroni adjustment is applied, these results will still be significant at 0.05/2 level of statistical significance.

5) The analyses do not account for overall gender composition of the pool of authors. If we only examine the 2-author papers and assume that each author is unique, there are 2*1000 + 581 + 447 = 3028 male authors compared to 2*377 + 581 + 447 = 1782 female authors. (Please also consult work by Sugimoto and Larivière for systematic analyses of male and female authorship patterns.) Is it really therefore surprising that more papers are led by male authors? The authors refer to the proportion of women trainees (citing reference 19) but provide no citation to support their implication that first authors may reasonably be assumed to be exclusively trainees.

We agree that the number of authors can potentially confound our analyses. We accounted for this variable by stratifying the analyses into 2 categories: two- vs. more than two-author publications. In addition, the results of the goodness-of-fit test suggest that based on the observed distribution we reject the null hypothesis of equal split between male- vs. female first author publications at 0.05 level of statistical significance.

6) The analyses do not account for the same person or people authoring multiple papers. Is this possible in this data set? If yes, this potential clustering effect could have been accounted for had they used a multilevel model for their analyses. See also point 3 above.

We did not find instances of the same author combination authoring multiple papers.

7) Changes in authorship credit allocation over time are accounted for in a very crude way by splitting the data set into two sets, dividing at 2007. This is not an appropriate way to account for change over time, and only adds to the problem of multiple hypothesis testing. See also point 3 above.

In our new analyses, we explored the relationship between the logit of gender bias and year of publication. The results suggest that the logit of probability decreases almost linearly until about year 2007-2008, after which it is rather stable, see Figure 2. Therefore, Year of Publication was modelled in two different ways: as a continuous and a categorical variable in two different models. In the paper, we report the estimated odds ratio of gender bias per additional year and comparing 2007+ vs 1995-2006.

8) The authors have split the dataset into articles with 2 equally-contributing authors and those with more than 2. Like the time-based split, this is problematic. Do the authors have any basis for assuming that there is a difference between these types of credit-sharing such that they must be analyzed separately?

In the biological sciences first authorship is considered the most important position. The importance of contributions declines with author order such that the second is generally considered to have contributed more than the third. The last position is usually reserved for the senior author. We focused only on multiple authors sharing the first author position. We felt that first author designation decisions might be done differently depending on whether there are only 2 authors versus if there are more than two authors with different gender composition. However, since there were many more two author publications N = 2,348 vs. more than two-author publications, n = 491, we could not adjust for this variable as a continuous predictor in the model. Therefore, instead we decided to divide the publications into 2 categories: two- vs. more than two authors and stratify our analyses by this variable.

9) In the Results section, the authors note that they looked at the possibility for alphabetical order in a subsample of 2109 papers. Why this subsample? How was this selected?

Frankly, the decision to include alphabetical information was done as the two authors were collecting data and so the subsample is dependent on date (meaning that once we decided to begin recording that information it was continues for the remainder of the 2109 papers we examined).

10) The authors report a confidence interval in their methods but do not apply it in any way in their presentation of results. More worrisome is that the authors have not accounted for the fact that confidence intervals rely on the assumption that the sample population (i.e., the data analyzed) was sampled randomly from the population. This was emphatically not the case in this study. For this reason, the confidence interval calculation does not apply. (A minor point in relation: I was able to reproduce their estimate of a confidence interval of 2.38 using the website they used by leaving the default setting of 50%, which was not the result in their study. This suggests a potential lack of understanding of how to use these calculations.)

We agree with the reviewer and have removed the sample size calculation since our sample was not truly random and the formalism requires this criteria for the calculation.

11) Four out of six continents lack sufficient sample size for it to be valid to look at distribution of shared authorship even descriptively. In addition, presenting these descriptive statistics in equally-sized pie charts inadvertently conveys that each continent has an equivalently-sized subsample in the data set.

We agree with the reviewer’s comment. Therefore, in our new analyses, we looked at the country as a three-category predictor: Europe, US and other. The results for these regions matches are comparable suggesting that the trends we are observing are occurring worldwide.

12) The data presented in this paper suggest that any gender bias in author order (which certainly appears to have been a legitimate issue in papers published 1995-2007) is no longer a problem in more recent papers. If this is true, this is excellent news and should be presented as such.

We agree that the trends are in the right direction and that this excellent news. However, since the effect of publication on a scientific career can be lifelong we worry that imbalances that occurred even a decade ago would be felt in the present as these individuals progressed through promotion, etc. We have rewritten the Discussion section to make the conclusion clearer. However, the presented results should be interpreted in light of the limitations of our approach, which we also state in the Discussion section.

[Editors' note: further revisions were requested prior to acceptance, as described below.]

Thank you for submitting the revised version of your manuscript "Disequilibrium in Gender Ratios among Authors who Contributed Equally". The revised manuscript has been reviewed again by the referee who had the most specific concerns about the statistical analyses employed in the original version, and while they welcome many of the changes, they still have a number of concerns that you will need to address (please see below). There are also several editorial points I would like you to address (again, please see below). Although this might seem like a long list (24 points), most of them should be straightforward to address.Reviewer #2:The authors have nicely addressed some of my concerns (in particular the statistical analyses are improved-the authors are to be commended for their work to improve the robustness of their conclusions) but some concerns remain, and some minor new concerns arose pertaining to the new analyses. My main concerns remaining have to do with how the statistical analyses are presented.1) First, for their new analyses, the authors have named their primary outcome variable "gender bias" and defined it as follows: "gender bias in first authorship was defined as first male author when all the authors contributed equally." This is problematic and does not align with how gender bias is defined in other literature. In an ideal world, there would be no gender bias, meaning that the optimal value of a variable named "gender bias" would be zero. Yet, presumably, the authors are not arguing that no man should ever be listed first under conditions of shared first authorship. By naming this variable "gender bias," this is what is currently implied. It would be more understandable to readers (and more in keeping with how this is usually done in studies of sex and gender in humans) to define this variable as something like, 'male author listed first' or perhaps 'm* authorship' in keeping with their notation used elsewhere in the paper. This makes it more intuitive to the reader that the ideal level for this variable is 0.5, not 0.0.

Change as requested. Interpretation is still discussed as bias if realized occurs greater than expected by pure chance.

2) Second, it is standard practice to provide a table in the results showing odds ratios and confidence intervals (along with p-values, if one believes those are important, which the authors seem to) for each variable in the model. This would help readers who are familiar with these kinds of statistics more rapidly grasp the findings.

Statistical information described above is provided in source files associated with figures and also presented where applicable in the text and figure legends.

I list these two points first and second because being clearer about how the outcome variable is defined and the odds ratios for each variable in the model could help solve some the following problem.3) In the Results section, the authors state, "The Odds Ratio of Gender Bias in First Authorship (95% Confidence Interval) using year as a continuous variable was 0.958 (0.931-0.986) with p < 0.01, indicating a significant preference for males in the first position considering all publications from 1995-2017." The statistics presented do not support the authors' statement here. Odds ratios under 1 mean that the outcome is less likely than the reference, not more likely. In other words, the odds ratio presented in this sentence indicates that the variable "gender bias" is less likely than the reference. The authors may want to check with their statistical consultants on this, because based on the raw data presented a few lines up (56% male author first vs. 44% female author first in the n=971 mixed-gender subsample) I would not be surprised to see an odds ratio and 95% confidence interval of 0.958 (0.931-0.986) for female first authorship in that subsample.

Yes, this was a miscommunication – since the outcome variable is “bias” defined as first male author among two-, equally contributing author papers, an odds ratio below 1 indicates that with each year the odds of gender bias go down by an estimated 4.2% (95% CI: from 1.4% to 6.9% decline in odds per year). This estimate is adjusted for country of origin.

4) Re: the alphabetical subset. The authors have added in the manuscript, "This number differs from the larger data set used in the analysis because we began to record alphabetical ordering after the study had begun when we realized this could be an important variable." I appreciate that they have explained why this happened. However, for this to be a complete, robust study, why don't they just quickly go back and record alphabetical or non-alphabetical ordering for the relatively small proportion of early papers (about one quarter of their sample? The fact that they didn't do it at the time doesn't preclude them from doing it now. This is particularly important given the journal's requirement to support replicability. Nowhere in the materials is it clear which papers in the dataset are part of the 2109 for which alphabetical or non-alphabetical order was recorded and which are part of the 787 for which such data were not collected at first glance.

Done. We went through the entire database and collected author alphabetical information for all entries. Updated Supplementary file 1 has all the alphabetical information. Numbers in the text were adjusted accordingly.

5) More details are needed regarding the data and analyses to ensure replicability. What statistical software (including version) was used? Typically, when writing about statistical analyses, we include the name of the software (e.g., SAS, Stata, SPSS, etc.) or statistical programming environment (R) and version number. This is because the same analyses run in different software could produce slightly different results.

We added a reference to the statistical software program that was used to perform the analyses (STATA version 15, StataCorp. 2017. Stata Statistical Software: Release 15. College Station, TX: StataCorp LLC).

6) Related to the above, is it possible to provide the script or code as an appendix along with the data? This is good practice for ensuring replicability.

The log file and the STATA do.file are now included as source files.

7) Results section: The authors state, "Comparing the expected and observed gender ratios yielded a Chi-square statistic of p <0.00.1" A p-value is not a Chi-squared statistic. A Chi-squared statistic would look something like Chi-squared(degrees of freedom)=number like 2.7 or 0.3 or 34.2 or 976.

Chi-squared statistic added. The Chi-squared statistic with 1 degree of freedom is 15.1.

8) Results section: "estimated 4%" should say "estimated 4% per year" and "1 to 7% decrease" should read "1 to 7% decrease per year."

Done as suggested.

9) The source data for Figure 2 doesn't appear to match the Figure. The Figure looks like it was plotted using year of publication as a quasi-continuous variable (i.e., there's one point for each year in the data set) while the source data shows year as a categorized variable (pre-2007 and 2007+) along with country/group of countries as a categorized variable.

We corrected the link to Figure 2–source data 1. Previously linked file was in error and should have been linked to Figure 5. Figure 5– source data 1are predicted probabilities from the generalized linear population average model with Binomial distribution, logit link, and robust variance estimate estimated using Generalized Estimating Equations (GEE) with bias as the outcome and publication year and country as predictors. In this model “bias” is defined as first male author among two equally contributing authors of both genders.

10) Point 11 noted, "Four out of six continents lack sufficient sample size for it to be valid to look at distribution of shared authorship even descriptively. In addition, presenting these descriptive statistics in equally-sized pie charts inadvertently conveys that each continent has an equivalently-sized subsample in the data set." The authors responded, "We agree with the reviewer's comment. Therefore, in our new analyses, we looked at the country as a three-category predictor: Europe, US and other. The results for these regions match are comparable suggesting that the trends we are observing are occurring worldwide." The analysis in the revision has addressed this very well and we now have a very nice Figure 5. However, Figure 4 continues to present 6 continents. This means that Figure 4 continues to have the problem of inadvertently conveying equivalency, and, in addition, it now fails to reflect the analyses conducted. I recommend simply removing Figure 4.

Figure 4 removed as recommended by the reviewer.

11) The original review stated: "16. If grammatically possible, please refer to 'women' or 'female authors' rather than 'females', and 'men' or 'male authors' rather than 'males.'" The authors' response to this was: "Response: Agree. Done." Yet, there are 22 instances in the manuscript of using 'males' or 'females' when it would be completely grammatically possible to refer to 'male authors,' 'men,' 'female authors,' or 'women.' For example, the second sentence in the revised abstract begins, "For mixed gender pairs males were…" A correct rewording of this line would be, "For mixed-gender pairs, male authors were…" It is acceptable to refer to mice as 'males' and 'females' but it is not acceptable to refer to humans in this way in most English-speaking countries, nor in scientific literature describing studies that involve humans as subjects, objects, or participants (e.g., in clinical medicine, social sciences, etc.) Please correct all 22 uses of 'males' and 'females'.

Sorry about this. It was an oversight on our part as we thought we had changed all the ‘males’ and ‘females’. In the revised version all have been corrected.

12) Point 16 in the original review noted that in the original Introduction, the authors stated that, "Analysis of articles in 5 medical journals showed that whereas papers listing equal contributions comprised less than 1% of publications in 2000, by 2009 this trend had increased to 3.6-8.6% depending on the journal (8)." In fact, the paper referenced by Akhabue and Lautenbach reports rates from 1.0-8.6% in 2009 in the top 5 general medicine journals. The 1.0% rate is from the BMJ. This means that the rate of co-first authorship in 2009 was 1.0-8.6%, not 3.6-8.6% as claimed by the authors. The change that they made in their revision (removing the reference to the 5 top general medicine journals) has not fixed the problem that their paper makes a claim unsupported by the citation they are using. They need to change 3.6% to 1.0% in order to be accurate.

Done as suggested. Text now states 1.0-8.6%.

13) In the file with file name Table 1–source data 1 what does the grey colouring on some rows mean? This should be specified so that anyone aiming to build on or replicate this work can understand the data file.

Grey lines indicate the papers for which gender identification was not able to be determined, this has been indicated in the legend.

14) Please consider changing the title to the following:“Gender Inequalities among Authors who Contributed Equally”

Done.

15) Please consider revising the abstract to read as follows:“We analyzed 2898 scientific papers published in the period 1995-2017 in which two or more authors shared the first author position. For papers in which the first and second authors made equal contributions, mixed gender combinations were most frequent, followed by male-male and then female-female combinations. For the mixed gender combinations, there were more male authors than female authors in the first position, although the disparity was less in the second decade of the period studied. For papers in which three or more authors made equal contributions, there were more male authors than female authors in the first position, and more all-male than all-female combinations. We also show that the disequilibrium in gender ratios among authors who made equal contributions is not consistent with random or alphabetical ordering of these authors. These results raise concerns about female authors not receiving their fair share of credit for scientific papers, and suggest a need for journals to request clarity on the method used to decide author order among those who made equal contributions.”

Done, as suggested. We inserted the suggested abstract wording verbatim except for the phrase ‘disequilibrium in gender ratios’ since point 16 below recommends changing this wording. We also modified some ‘male’ to ‘male authors’ as recommended by reviewer 2, point 11.

16a) Please consider replacing the term "disequilibrium in gender ratios" with the term "gender inequalities".

Done as suggested.

16b) Please consider replacing the terms associations, author associations etc. with the terms combinations, author combinations.

Done as suggested.

17) Please reword the phrase "the person who does the actual work"

Done.

18) Re the sentence that starts: "It is conceivable that some of the disequilibrium in gender ratio in the earlier years…", is it worth mentioning here or somewhere else in the manuscript that, irrespective of the gender breakdown of the science workforce, when considering just those manuscripts with two first authors, one would expect mf to equal fm?

Sentence was rephrased.

19) Please reword the following sentence to be more precise and/or avoid the word "predominated" (and similar words): "We observed that male-only pairings predominated in author combinations of two or more authors."

Sentence was rephrased.

20) The sentence "The frequency of multi-author equal contributions dropped rapidly for associations of more than three authors but we observed at least two groupings of 11 authors [Bonham and Stefan, 2017; van den Besselaar, Sandstrom, 2017]." suggest that Bonham and Stefan, 2017 and van den Besselaar and Sandstrom, 2017 have 11 authors, but this is not the case: please clarify.

The references have been corrected. New references Massen et al., 2017 and Bendals et al., 2018 each lists 11 authors sharing the first position.

21) Please move the passage about the limitations caused by the use of Google Scholar ("The approach to search […] in the text of the paper.") to the Materials and methods section, and add a short sentence to the main text stating that this limitation is discussed in the Materials and methods section.

Done.

22) Please expand the caption for Figure 2 to better explain to the reader what is shown in this Figure. The revised caption should include a title sentence (in bold). Also, please explain in the caption what the ideal level of this logit function would be. Please also define p(bias).

Done.

23) Please expand the caption for Figure 3 to better explain to the reader what is shown in this Figure. The revised caption should include a title sentence (in bold).

Done.

24) Please expand the caption for Figure 4 to better explain to the reader what is shown in this Figure. The revised caption should include a title sentence (in bold). Please also explain how these values are predicted.

Figure 4 was removed as per the suggestion of reviewer 2.

[Editors' note: further revisions were requested prior to acceptance, as described below.]

Reviewer #2:Thank you very much for the opportunity to re-review this paper. The authors have responded extremely well to nearly all my comments, and I believe the paper is substantially improved. There are just a few small corrections remaining:1. Regarding this passage in the Discussion: “We noted differences between journals in the proportion of pairings…” […]. It's up to the authors, of course, but I think they are underselling their results at this point by calling their findings "preliminary". Also, software for gender identification often uses names as well as, or instead of, images. I would suggest changing, "Hence, our findings should be considered preliminary until confirmed by subsequent studies, which may be able to analyze a larger number of publications across many disciplines through automated searches linked to gender image recognition software," to something like, "Our findings should be complemented by subsequent studies, which may be able to analyze a larger number of publications across many disciplines through automated searches linked to gender recognition software."2. In discussing their findings in the context of the broader literature, the authors may wish to refer to a similar study published in JAMA, in February 2018, that came across my radar recently: https://www.ncbi.nlm.nih.gov/pmc/articles/PMC5838607/

We appreciate the comment about underselling the results and amended the text as suggested. We also incorporated the recent study above within this context, since this study had similar results using both out method and some additional databases. The relevant text now reads:

“In this regard, a recent study that examined papers from 2005-2014 with co-first authors of different genders using online databases in addition to our method of website images, found that while there was no difference in female versus male authors in the first position among basic science journals, consistent with our results (Aakhus et al., 2018). However, they found that female co-first authors were less represented in the first position among clinical journal. As such, our finding that a disequilibrium exists…”